# The Use of Point-of-Care Ultrasound (POCUS) in the Diagnosis of Deep Vein Thrombosis

**DOI:** 10.3390/jcm10173903

**Published:** 2021-08-30

**Authors:** Dimitrios Varrias, Leonidas Palaiodimos, Prasanth Balasubramanian, Christian A Barrera, Peter Nauka, Angelos Arfaras Melainis, Christian Zamora, Phaedon Zavras, Marzio Napolitano, Perminder Gulani, George Ntaios, Robert T. Faillace, Benjamin Galen

**Affiliations:** 1Department of Medicine, Jacobi Medical Center, 1400 Pelham Parkway South, 3N1, Bronx, NY 10461, USA; balasubp2@nychhc.org (P.B.); barrerac@nychhc.org (C.A.B.); arfarasa@nychhc.org (A.A.M.); zamorac@nycch.org (C.Z.); zavrasf@nychhc.org (P.Z.); naplitam@nycchc.org (M.N.); perminder.gulani@nychhc.org (P.G.); robert.faillace@nychhc.org (R.T.F.); 2Albert Einstein College of Medicine, Bronx, NY 10461, USA; penauka@montefiore.org (P.N.); bgalen@montefiore.org (B.G.); 3Department of Medicine, Montefiore Medical Center, Bronx, NY 10467, USA; 4Department of Medicine, School of Health Sciences, University of Thessaly, 41500 Larissa, Greece; g.ntaios@med.uth.gr

**Keywords:** POCUS, ultrasound, DVT, VTE, sonography, point-of-care ultrasound, deep vein thrombosis, venous thromboembolism

## Abstract

Acute lower extremity proximal deep venous thrombosis (DVT) requires accurate diagnosis and treatment in order to prevent embolization and other complications. Point-of-care ultrasound (POCUS), a clinician performed, and clinician interpreted bedside ultrasound examination has been increasingly used for DVT evaluation mainly in the urgent and critical care setting, but also in the ambulatory clinics and the medical wards. Studies have demonstrated that POCUS has excellent diagnostic accuracy for acute proximal DVT when performed by well-trained users. However, there is significant heterogeneity among studies on the necessary extent of training and universally acceptable standardized education protocols are needed. In this review, we summarize the evidence that supports the use of POCUS to diagnose acute proximal DVT and focus on methodology and current technology, sensitivity and specificity, pre-test probability and the role of D-dimer, time and resources, education, limitations, and future directions.

## 1. Introduction

Acute lower extremity proximal deep venous thrombosis (DVT) is a serious vascular condition with an annual incidence of 0.1% in adults [1]. Accurate diagnosis and treatment of acute DVT is crucial in order to prevent embolization and other complications. Mortality from pulmonary embolization, a potentially fatal complication of DVT, could be up to 30% if left untreated [2]. Although the gold standard for the diagnosis of DVT is contrast venography, duplex ultrasonography performed by radiology or vascular laboratories has become the standard of care due to its widespread availability, cost-effectiveness, lack of radiation, lack of intravenous contrast, and patient comfort [3]. Point-of-care ultrasound (POCUS), a clinician-performed and clinician-interpreted bedside ultrasound examination, has been increasingly used in the emergency department (ED), the intensive care unit (ICU), medical wards and in the outpatient setting for evaluation of the proximal lower extremity venous system [4,5,6,7]. Studies have found that POCUS may have comparable diagnostic accuracy to radiology or vascular lab-performed duplex ultrasonography for the detection of proximal lower extremity DVT, which makes it a very useful tool in routine clinical practice [4,5]. The American College of Emergency Physicians has supported the use of POCUS by trained physicians to evaluate for DVT since the 1990s [8], but it was not until 2017 that DVT was added in the list of twelve core ultrasound applications for emergency medicine physicians [9]. Interest in using POCUS for diagnosing DVT has grown substantially, not only in emergency medicine and critical care, but also in internal medicine and hospital medicine.

We conducted a narrative review of studies that evaluated the use of POCUS in the diagnosis of DVT. In our review, we summarize the evidence that supports the use of POCUS to diagnose DVT, including methodology and current technology, sensitivity and specificity, pre-test probability and the role of D-dimer, time and resources, education, and limitations.

## 2. Methods

Studies were considered eligible for consideration in this narrative review if they were reporting on use of POCUS on the diagnosis and management of DVT. In specific, eligible articles were those that included information for any of the six areas of interest regarding the use of POCUS and diagnosis of DVT: (1) methodology and current technology, (2) sensitivity and specificity, (3) pre-test probability and the role of D-dimer, (4) time and resources, (5) education, and (6) limitations.

The MEDLINE and Embase databases were searched from database inception to 28 February 2021. A combination of free-text words and MeSH subheadings were used, including the terms “POCUS”, “portable ultrasound”, “CUS”, “ultrasound”, “compression ultrasound”, “DVT”, “VTE”, “deep vein thrombosis”, and “venous thromboembolism”.

## 3. Equipment and Current Technology

Ultrasound machines used in existing literature can range from pocket-sized handheld devices to more sophisticated trolley or cart mounted machines with varying cost, ranging between 1500 and greater than 30,000 US dollars [10,11,12]. The linear transducer is used in the venous mode with depth ranges between 1–4 cm and frequency set around 7.5 Mhz [13,14] (Figure 1). Most handheld manufactures sell a probe that can be used for vascular applications and many of these devices include a probe with multiple applications.

## 4. Patient Preparation

The patient should be positioned supine and the head should be elevated to 30° preferably, which can help blood pooling in the veins of the lower extremities and aid in visualization of the vasculature [15]. Then the examiner should externally rotate the patient’s hip and bend the knee slightly into what is known as the “frog leg” position [15]. This is the most popular position because it enlarges the femoral veins and brings it closer to the field of vision for the plane of the ultrasound probe. In addition, the “frog leg” position allows the examiner to scan the inguinal region and the popliteal fossa without having to reposition the patient [16]. When possible, prone position can help in examining the popliteal veins [17]. The examiner is typically most comfortable standing at the patient’s side, ipsilateral to the extremity being evaluated. When using a cart-based ultrasound machine, it should be positioned within the examiner’s reach at the head of the bed. The bed height should be elevated for examiner comfort [12]. The patient preparation and positioning are depicted in Figure 2 and Appendix A.

## 5. Protocols

There is variability in the definition of POCUS protocols to evaluate for DVT and their names, as well as the levels they include, can be a source of confusion for learners. The two main types of POCUS examinations that have been studied for the diagnosis of DVT are called “2-point” and “3-point” exams, which is somewhat of a misnomer because they both involve scanning more levels or segments of vein than that. Even within 3-point protocols, there is variability in the levels of proximal lower extremity vein included. The term “compression ultrasound” is used historically to refer to bedside ultrasound in B-mode without Doppler. This involves inspection for echogenic thrombus and if a thrombus is not seen, testing for compressibility of the visualized segment of vein with the probe in the transverse orientation [12]. For the purpose of this review, the term POCUS is used and unless otherwise specified refers to B-mode inspection for thrombus and the compression ultrasound technique.

The “2-point” POCUS technique evaluates the common femoral vein (CFV) and the popliteal vein (PV) [18,19,20,21]. In the 2-point examination, the common femoral vein is assessed from the inguinal ligament until it becomes the femoral vein (FV) (historically named the superficial femoral vein). This includes the common femoral vein–greater saphenous vein (CFV-GSV) junction. The PV is assessed starting from where it is parallel to the popliteal artery and following it until the trifurcation [14]. In the “3-point” technique, the CFV and PV are evaluated as in the 2-point examination, but further scanning of the FV in the proximal thigh is undertaken [11,22,23]. The 3-point compression protocol begins at the level of CFV and includes the CFV-GSV junction, the proximal FV, mid-distal FV, and PV (Figure 2). A graphic representation of the aforementioned vein levels typical for a 3-point POCUS examination is shown in Figure 3. The use of color Doppler or spectral Doppler are advanced techniques when it comes to point of care setting but can be very useful in determining the degree of occlusion when a DVT is found [11]. Determining the chronicity of a DVT is also typically reserved for vascular lab and diagnostic radiology, not POCUS evaluation.

In all POCUS techniques, the diagnosis of DVT is made by visualizing echogenic thrombus (Figure 4 and Figure 5) or if an area of vein is not able to be fully compressed. Acute venous thrombosis often results in non-compressibility of a vein before an echogenic clot can be visualized. When testing for compression of a vein, the examiner should apply enough pressure so that the pulsatile artery nearby compresses slightly. The pressure should be applied rapidly and perpendicular to the vein with the probe in transverse orientation, like a piston. Weak or off-axis compression can result in a false positive result [12]. Rarely, too much pressure can result in a false negative result. The 3-point technique along with POCUS clips of the various vein levels without evidence of clot are demonstrated in Appendix A. POCUS clips demonstrating DVT in various levels with or without visible thrombus are presented in Appendix A.

## 6. Sensitivity and Specificity

Several studies have evaluated the sensitivity and specificity of emergency physician-performed POCUS to evaluate for DVT. In 1997, Jolly et al. assessed two emergency physicians’ accuracy in performing duplex ultrasound as compared to the vascular laboratory studies. The physicians were trained by the vascular laboratory technicians and were required to complete 25 to 30 technically adequate examinations prior to independent scanning. The investigators reported a sensitivity of 100%. However, we should acknowledge that this study involved well-prepared physicians performing lengthy complete duplex examinations and not the 2-point or 3-point POCUS techniques [24]. In 2000, Blaivas et al. performed examinations using a 2-point compression ultrasound at FV and PV. These POCUS results had a high correlation with vascular laboratory studies, giving a kappa of 0.9 and a 98% agreement (95% CI 95.4–100%) [25]. Magazzini et al. similarly found a sensitivity of 100% and specificity of 98.4% of POCUS for DVT performed by emergency physicians [26]. In 2004, Jang et al. studied the ability of emergency medicine residents to perform a compression-only examination of the entire proximal leg including the popliteal fossa. Resident-performed scans were 100% sensitive and 91.8% specific in diagnosing proximal DVT as compared to imaging performed by vascular technicians. Kory et al. included critical care fellows and attending physicians who performed a 2-point POCUS protocol and found comparably accuracy with complete vascular laboratory study [27]. In a study performed by Pedraza et al., 3-point POCUS examination performed by emergency physicians had a sensitivity of 93.2% (95% CI 83.8–97.3%) and a specificity of 90% (95% CI 78.6–95.7%) which is comparable to existing data from the radiology-performed duplex ultrasonography [28]. Metanalyses on the topic consistently showed that POCUS had a sensitivity between 90–95% and specificity between 91–98% [4,5,29]. Fischer et al. performed the HOCUS-POCUS study, where they evaluated the sensitivity, specificity, and predictive values of 3-point POCUS technique performed by hospitalists in non-ICU hospitalized patients compared to vascular lab ultrasound [11]. Prior to the study, hospitalists attended a 2-h didactic and hands-on training and completed ten normal DVT studies on standardized patients. HOCUS-POCUS revealed a sensitivity of 100% and specificity of 95.8% with a positive predictive value of 61.5% and a negative predictive value of 100%.

On the other hand, despite promising results showing comparable accuracy between radiology- or vascular lab-performed techniques and POCUS, a prospective cohort study from Caronia et al. reported that a 2-point POCUS technique performed by internal medicine residents was not adequate for detecting all proximal DVT. While residents were able to detect all common femoral and popliteal clots, isolated femoral clots were missed as this was an area not included in the 2 point protocol used in this study [30].

In terms of comparing different protocols/techniques (2-point vs. 3-point) and in contrast to the findings of Coronia et al. that included internal medicine residents [30], a meta-analysis performed by Lee et al. demonstrated that 2-point and 3-point POCUS were both excellent methods for the diagnosis of DVT with similar sensitivity and specificity in different settings with different performers [29].

Overall, the existing studies have shown that POCUS has excellent diagnostic accuracy for evaluation of acute proximal DVT. However, it should be emphasized that these studies have significant heterogeneity as far as the setting (ICU, ED, medical ward), as well as the preceding training of the clinicians performing POCUS (resident vs. attending). Therefore, extrapolating these findings to different users with varied training must be undertaken with caution.

## 7. Pre-Test Probability and D-Dimer

Crucial to the evaluation of suspected DVT is an accurate determination of pre-test probability for a clot. Widely accepted guidelines recommended the use of validated scores and high sensitivity D-dimer to evaluate the likelihood of a diagnosis of DVT in patients presenting with suggestive symptoms [31].

Use of algorithms incorporating pre-test probability assessment with a sensitive D-dimer test have been shown to reduce the number of imaging studies performed [32,33]. A scoring system from Wells et al. is the most commonly used in clinical practice where a score ≥2 indicated a high pre-test probability of DVT [33]. After an unlikely pretest probability of DVT, a negative D-dimer test is adequate to safely rule-out DVT and venous ultrasound may not be appropriate for those individuals [33]. Ultrasound to diagnose DVT is appropriate for patients with a likely pretest probability or an unlikely pretest probability with a positive D-dimer test [32,33,34]. It should be emphasized that the above recommendations were designed for radiology or vascular lab-performed ultrasonography but can provide a helpful tool in the hands of clinicians using POCUS regarding risk classification. Pedraza et al. demonstrated a good diagnostic accuracy of three-point POCUS for DVT detection in the ED when integrated in a diagnostic protocol that included Wells’ criteria and a D-dimer test for patients with suspected DVT. A total of 109 patients underwent a 3-point ultrasound in the ED by emergency physicians and a second ultrasound performed by radiology. Bedside three-point POCUS preceded by application of Wells’ criteria and a D-dimer test as indicated had a sensitivity of 93.2% (95% CI 83.8–97.3%) and a specificity of 90.0% (95% CI 78.6–95.7%), with an accuracy of 91.7% (95% CI 85–95.6%) [28]. Better diagnostic outcomes using venous and lung POCUS in combination with the Wells criteria were also validated on a study performed by Nazerian et al. [35].

## 8. Time and Resources

The main advantage to POCUS in the evaluation for DVT is that it can be performed immediately at the beside by the clinician using a device already readily available in the ED, the ICU or on the medical ward. Although there are studies calculating the amount of money saved by implementing POCUS in various settings, we did not find any studies that evaluated the cost-effectiveness of POCUS in the diagnosis of DVT [36,37]. As far as turnaround time is concerned, a meta-analysis performed by Pomero et al. revealed that POCUS, as an alternative to radiology or vascular lab-performed venous ultrasound, may have an important role in current healthcare especially when these services are not available on a 24-h basis [4]. In another study, POCUS appeared to have practical utility in the context of a busy emergency department setting since the average time from request to completion of study was within the first 15 min [4]. When the equipment is available to trained personnel, the median time taken to perform a POCUS exam is estimated between 3–5 min [25,28,38]. Subsequently, Theodoro et al., sought to address the question of how these studies influenced the time-to-department disposition (discharge from ER or hospital admission) when patients presented with signs or symptoms concerning a DVT. The time the emergency physician made the diagnosis was used as the emergency physician disposition time and the time the report was received from the radiology department was used as the radiology disposition time. The mean time from triage to disposition was 95 min for patients who had their ultrasound performed by the emergency physician, whereas the mean time from triage to disposition was 220 min for patients who had their study performed by the radiology department. There was a high correlation (kappa = 0.9) and 99% agreement of the emergency physician–performed studies with radiology-performed ones [21]. In the HOCUS-POCUS study, the median time from order to POCUS completion by hospitalists on the medical ward was 5.8 h versus 11.5 h median time from order until the radiology report was finalized (*p* = 0.001) [11]. When investigating the effect of POCUS on length of stay, Chen et al. showed that median length of stay in the ED of patients seen by the emergency medicine residents decreased from 80 (IQR: 42–155) to 75 (IQR: 39–146) minutes, while the return rates to ED within 24 h and 72 h decreased slightly from 1.37% to 1.31% and 2.44% to 2.42%, respectively [10]. On the other hand, when POCUS is not available, patients may require hospitalization and/or empiric anticoagulation, until DVT is definitively diagnosed or excluded. Vascular lab- or radiology-performed DVT studies are not universally available at all healthcare settings and technologists may only be available during weekday daytime hours, which delays diagnosis [39,40]. Another important point is that POCUS could be very useful in the emergent setting when resources like CTPA are available but contraindicated (pregnancy, severe renal failure and allergy to contrast) as described by Squizzato et al. [41]

## 9. Education

An important consideration when introducing POCUS in clinical practice is determining the necessary training and experience. The American College of Emergency Physicians guidelines suggest performance of the POCUS for clinical decision making after completion of a two-day course and 25–50 quality assurance reviewed studies [9,42,43]. There is a lot of variability in educational material and curriculum used regarding education on POCUS techniques in general, but also when it comes to the diagnosis of DVT. Chen et al. showed great results with a combination of theoretical lectures, followed by hands-on challenges and frequent knowledge tests [10]. Although a brief training in this technique could be attractive for emergency physicians, the implications of inadequate training are considerable, and comprise errors of omission (not treating DVT when it is falsely excluded) and errors of commission (starting anticoagulant therapy when DVT is falsely confirmed) are possible [4]. Blaivas et al. cautioned that “10 min and you are ready to go” is not quite sufficient for training; however, he echoed that with proper training, emergency physicians can accurately diagnose DVT in the emergency department [25]. In detail, Blaivas at al. showed that a training POCUS session of two hours of didactic education followed by three hours of hands-on learning in conjunction with prior experience in POCUS has high correlation with vascular laboratory studies, giving a kappa of 0.9 and a 98% agreement (95% CI 95.4–100%) [44]. Eight resident physicians that participated in the Jang et al. study received a 1 h lecture and a demonstration of the technique on two healthy volunteers resulting in excellent diagnostic accuracy. The investigators concluded that residents with limited training could quickly and accurately perform POCUS for DVT evaluation [22]. There are many discrepancies in the education curriculum, hence, Fox and Bertolio called for a more uniform and universal training of emergency physicians for the use of POCUS to diagnose DVT [45].

As pointed out by Andersen et al., diagnostic accuracy improves with hours of practical training, and studies incorporating continuous feedback on scans conducted during clinical patient encounters show superior results [46]. Another study found that resident physicians initially failed to demonstrate sufficient sensitivity in a cohort of critically-ill patients due to a high rate of missing femoral vein thrombosis (6/21, 28%) [30]. After a 2 h course in vascular POCUS, resident physicians showed substantial agreement with radiologists for the diagnosis of clinically-relevant DVT [30,47]. A meta-analysis conducted by Lee et al. revealed that the initial POCUS performer and training before the study were sources of heterogeneity. In particular, the pooled sensitivity was higher in studies including the attending emergency physician than in studies including only the resident physician. In addition, the pooled specificity was higher in studies that included POCUS training for DVT before the study [29]. In one of the first studies on the subject, physicians were trained by the vascular laboratory technicians and were required to complete 25 to 30 technically adequate examinations prior to performing studies in the emergency department resulting in 100% sensitivity and 75% specificity [24]. Choi et al. demonstrated a prominent decrease in length of stay in the ED (6.55 to 5.25 h) and revisit rate to the ED within one year (6.4% to 5.25%) after a systematic POCUS education program [48]. Magazzini et al. enrolled 399 patients to receive a study by one of two emergency department physicians, who each completed 30 h training, followed by a formal duplex ultrasound in a vascular laboratory within 24 to 48 h of discharge. Whereas this study was encouraging regarding the ability of emergency physicians to accurately perform lower extremity ultrasound for the diagnosis of DVT, its applicability and generalizability was questionable given the extensive training provided to those involved and the time needed to perform such a comprehensive examination [26].

Overall, there is evidence supporting the assertion that clinicians or other skilled staff can perform POCUS to diagnose or rule out proximal DVT with high diagnostic accuracy after training [46]. However, it is not clear how extensive the training should be and what it should entail. More studies are needed to further evaluate this domain.

## 10. Limitations

The pooled proportions of the false-negative rate of the 2-point and 3-point POCUS have been estimated around 4% and are similar between the 2 protocols [49,50]. A false negative evaluation for DVT would mean underdiagnosing thrombosis potentially leading to life-threatening pulmonary embolism. The pooled proportions of the false-negative rate of the 2-point POCUS and 3-point POCUS have been estimated around 4% and are similar between the two protocols [49,50]. Whether a full examination for distal DVT is appropriate or is still a debate, there are publications showing that 48% of patient with PE have contemporary isolated distal clots while this percentage for negative PE population is only 12% [51]. Unfortunately, these patients would be considered false negative while performing a standard POCUS protocol instead of a full CUS performed by a radiologist [52,53]. POCUS can also have false positives around 4% as reported by Fischer et al., which can lead to unnecessary anticoagulation treatment and risk of bleeding or other complications [11,54]. There are several potential causes of a false positive POCUS evaluation for DVT worth noting. Superficial thrombophlebitis can be mistaken for DVT, however the major difference is that superficial veins do not accompany arteries while deep veins do [55]. A Baker’s cyst appears as a circular anechoic structure in the popliteal fossa that can resemble a non-compressible vein (Appendix A) [56]. Enlarged inguinal lymph nodes can appear similar to a vein with a clot because of their oval structure, anechoic rim with hyperechoic center, but are easy to distinguish from a vein by scanning through to see that the structure is self-contained and even easier when the true CFV and arteries are found in a different fascial plane (Appendix A). Rarely, pseudoaneurysms and groin hematomas can be falsely diagnosed as DVT by non-experienced performers [56,57]. An important technical consideration for POCUS performers learning to evaluate for DVT is knowing how much to compress the vein. If the operator does not apply enough pressure or if the ultrasound probe is not perpendicular to the vein, this can lead to false positives [58]. Another cause of false-positive POCUS examination for DVT is the presence of “rouleaux formation” that is an accumulation of erythrocytes lying over the venous valves represented by spontaneously echogenic blood flow inside the vessel. While rouleaux is a common finding and usually does not have a clinical impact, it is important to highlight that this can also be frequently observed when there is a proximal venous obstruction, and thus a more proximal DVT must be ruled out. In contrast to a real condition of thrombosis, veins with rouleaux formation are compressible [59]. While POCUS can be immensely helpful in the rapid diagnosis of DVT at the bedside, equivocal findings should prompt further evaluation by radiology or a vascular lab-performed duplex ultrasound study.

## 11. Summary and Future Directions

Acute lower extremity proximal DVT is a common clinical entity frequently found in differential diagnosis in outpatient and inpatient settings that requires urgent evaluation. POCUS, which is widely used in the ED and ICU, has also been gaining popularity in primary care and hospital medicine due to increased training opportunities and reduced cost of equipment. POCUS has a well-demonstrated role in the rapid evaluation of suspected DVT in the emergency department and in critically ill patients, and thus there is increased interest in using POCUS to diagnose DVT on the medical/surgical wards as well as ambulatory clinics.

The 2-point and the 3-point exams are two widely used POCUS protocols for lower extremity proximal DVT evaluation. Although the latter requires scanning of additional segments of vein, both protocols can be performed within minutes by experienced users. The available literature supports that both protocols have excellent sensitivity and specificity when performed by trained clinicians. However, it should be recognized that the current evidence has been obtained from varied POCUS users with heterogeneous level of training and practicing in heterogeneous clinical settings: ED, ICU, medical ward). Therefore, given that the 3-point protocol provides additional evaluation of proximal and mid-distal FV without being significantly longer in duration, we conclude that the 3-point protocol should be considered over the 2-point one when evaluating stable patients. The 2-point protocol might be preferred in unstable patients as it takes less time. Importantly, neither of these POCUS protocols involves evaluation for distal (below knee) DVT, a less serious entity with lower risk of complications [60,61]. Therefore, when evaluation for distal DVT is indicated, a formal duplex ultrasound should be considered.

The extent of training that is considered adequate prior to incorporating POCUS in clinical practice for proximal DVT requires further study. There is significant heterogeneity in training level of the users in the available published studies, as well as variability in the curriculum offered by professional societies, such as the American College of Physicians, American Thoracic Society, and American College of Emergency Physicians. While excessive requirements could be a barrier for POCUS use, inadequate training may lead to a false sense of confidence and improper clinical decisions with potentially serious implications. Therefore, further research is needed to determine competency for POCUS users from all clinical fiends in the evaluation for proximal DVT. Future studies should examine the diagnostic accuracy of POCUS users who have received a standardized curriculum, not only early after training but also post-training so that quality assurance is assessed. Professional societies may need to work jointly towards establishing common guidelines on POCUS education and providing institutions, clinicians, residency programs, and medical schools with universally acceptable training protocols.

Another area of uncertainly is the role of the D-dimer test in the evaluation for proximal DVT using POCUS. While D-dimer is a widely used laboratory test in patients with low pre-test probability for DVT to facilitate deciding whether a formal duplex ultrasound is needed or not [62,63], its role on POCUS evaluation for DVT has only minimally been explored. Well-designed large studies are needed to evaluate whether the D-dimer test in conjunction to POCUS can increase diagnostic accuracy.

In conclusion, POCUS is a reliable diagnostic tool in the hands of well-trained clinicians suspecting acute proximal DVT in a variety of clinical settings. The decreasing equipment cost and increasing training opportunities have made POCUS more accessible giving the opportunity to clinicians to decrease time to evaluation and save resources. Both 2-point and 3-point exams can be completed within minutes and seem to have excellent diagnostic accuracy when performed by experienced users. The “frog-leg” patient position is preferred in both exams. Although POCUS has made well-trained users capable of excluding or diagnosing acute proximal DVT at the bedside, it should be emphasized that inadequate training can lead to misdiagnosis. Therefore, universally acceptable standardized training protocols are needed across medical and surgical specialties.

## Figures and Tables

**Figure 1 jcm-10-03903-f001:**
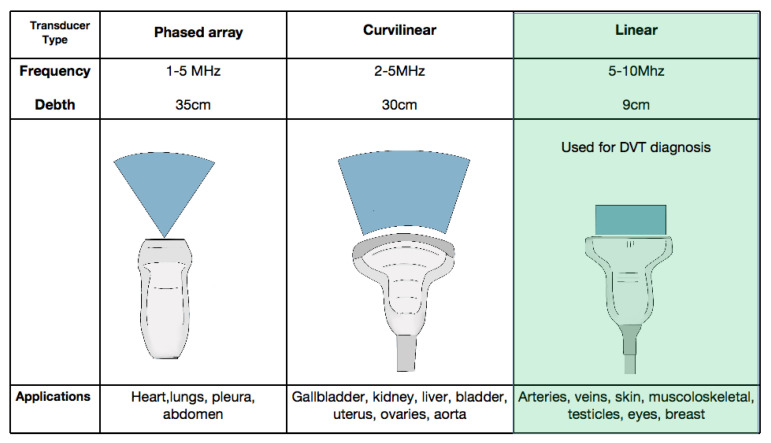
The three types of transducers used in point-of-care ultrasound (POCUS). The linear transducer (light green) is preferred for the evaluation of veins. DVT = deep vein thrombosis.

**Figure 2 jcm-10-03903-f002:**
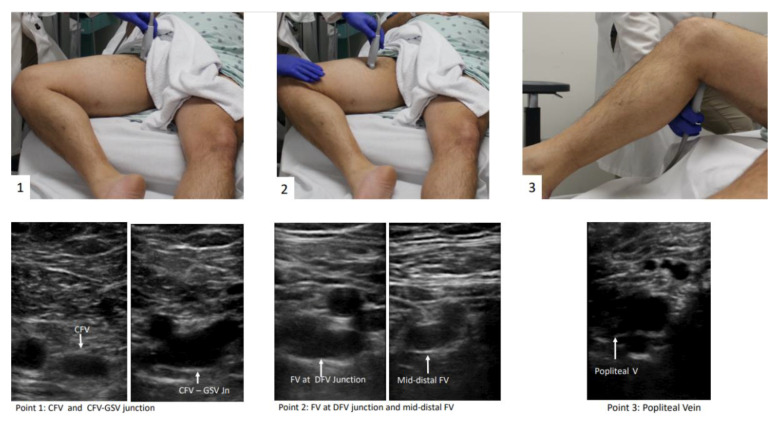
3-point POCUS technique for evaluation of acute proximal DVT in the right lower extremity. The “frog-leg” position is depicted. Veins were presented in all points before compression application. Abbreviations: Rt = right, CFV = common femoral vein, GSV = great saphenous vein, FV = femoral vein, DFV = deep femoral vein.

**Figure 3 jcm-10-03903-f003:**
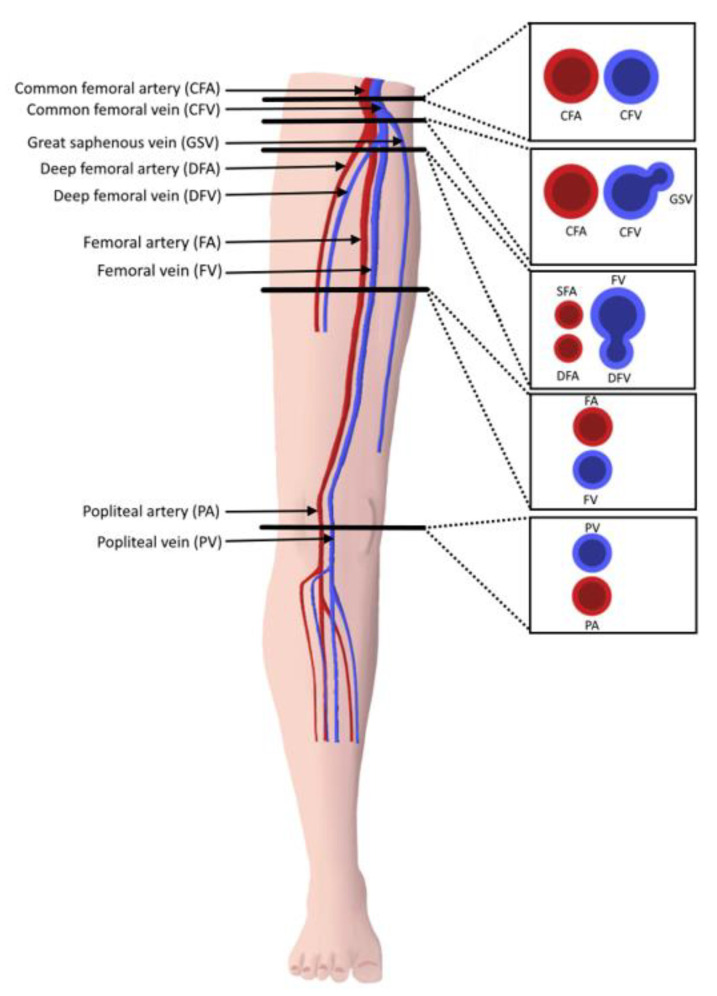
Cross-sectional anatomy of lower extremity proximal vasculature. Compressions should be performed at each of the marked points (3-point technique).

**Figure 4 jcm-10-03903-f004:**
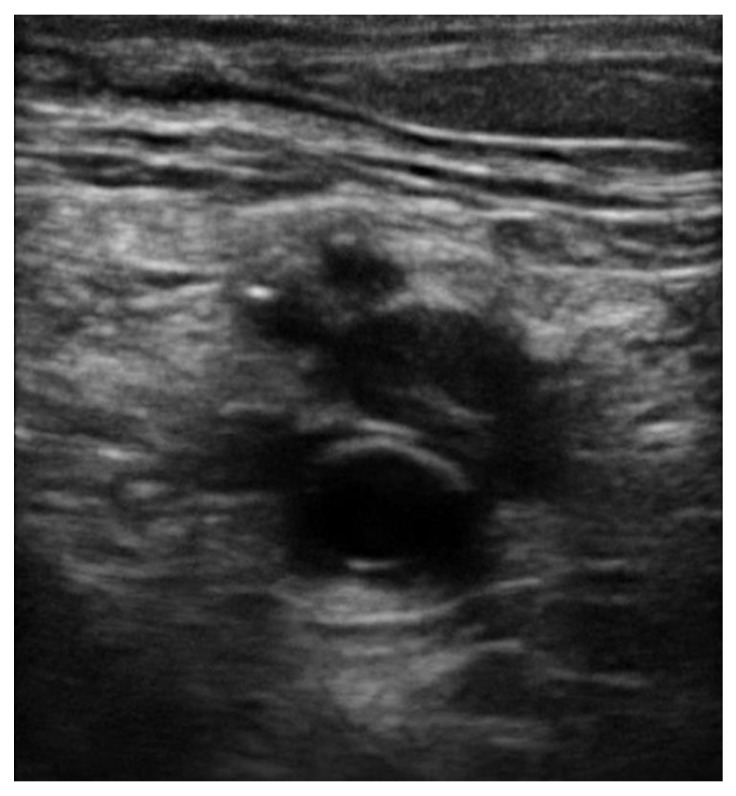
Echogenic thrombus is visible in the right popliteal vein (vein on the top of the artery).

**Figure 5 jcm-10-03903-f005:**
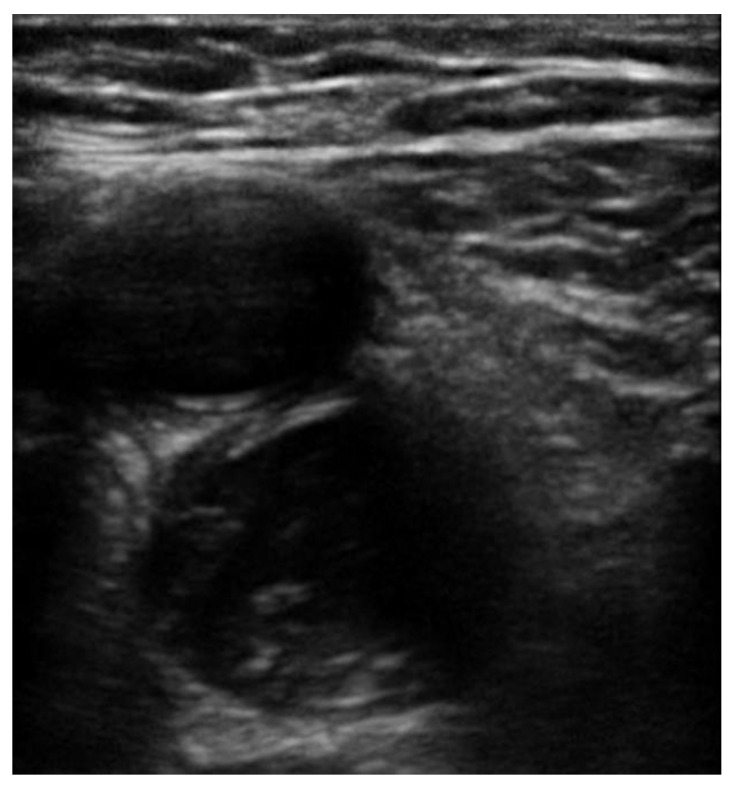
Echogenic thrombus is visible in the left distal femoral vein (artery on the top of the vein).

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
