# Peer review of "The Use of Point-of-Care Ultrasound (POCUS) in the Diagnosis of Deep Vein Thrombosis"

_jcm, 2021, doi:10.3390/jcm10173903_

Round 1
Reviewer 1 Report
This is a well-written review about the use of point of care ultrasound for detecting deep vein thrombosis. The article is well structured and written. Authors may consider the following suggestions:
- Line 36: authors should add "medical wards and in outpatient settings" to emergency department and intensive care units.
- Figure 1 should be deleted due to its lack of interest.
- Line 82: sometimes it is useful to ask the patient to lie face down to explore the popliteal vein (if possible).
- Line 114: Is the use of color doppler an advanced technique?. It is easy to perform and adds significant information to compression examination.
- Line 205. Authors should consider mentioning the following reference published by Nazerian et al. Acad emerg med. 2017; 24: 270-280. PMID:27859891 doi: 10.1111/acem.13130
- Line 240: Authors should consider to include some information about the usefulness of DVT detection in patients in which pulmonary embolism is suspected but CT scan has important risks for patients (pregnancy, contrast allergy, renal insufficiency...).
- Line 324. Authors should consider including a brief commentary about the limitations related to not performing distal veins examination like when radiologists perform a comprehensive study.
Author Response
Thank you so much for your valuable comments. Below we resolve every one in order.
- Line 36: authors should add "medical wards and in outpatient settings" to emergency department and intensive care units :
has been increasingly used in the emergency department (ED), the medical wards, the intensive care unit (ICU) and the outpatient setting for evaluation of the proximal lower extremity venous system [4-7].
- Figure 1 should be deleted due to its lack of interest. : Team believes that this figure could be useful for people that have rudimentary knowledge of the equipment but you are right that it is not useful for people with high expertise in the field. Since the paper covers a large spectrum in terms of depth, we decided to keep it
- Line 82: sometimes it is useful to ask the patient to lie face down to explore the popliteal vein (if possible): Thank you for that correction. Below you can find our addition to the manuscript along with the references to support this argument
When possible, having the patient lie on a prone position can be very helpful for scanning the popliteal veins. (Ref: https://doi.org/10.1016/j.ejvssr.2019.05.004)
- Line 114: Is the use of color doppler an advanced technique?. It is easy to perform and adds significant information to compression examination: Thank you for that correction. Below you can find our addition to the manuscript
The use of color doppler or spectral doppler are somehow advanced techniques to determine degree of occlusion when a DVT is found but at times difficult to be performed in the point of care setting [11]
- Line 205. Authors should consider mentioning the following reference published by Nazerian et al. Acad emerg med. 2017; 24: 270-280. PMID:27859891 doi: 10.1111/acem.13130 : Thank you very much for pointing that out. Will do.
- Line 240: Authors should consider to include some information about the usefulness of DVT detection in patients in which pulmonary embolism is suspected but CT scan has important risks for patients (pregnancy, contrast allergy, renal insufficiency...): Thank you for that correction. Below you can find our addition to the manuscript along with the references to support this argument
Another important point is that POCUS could be detrimental in the emergent setting when resources like CTPA are availbale but containdicated (pregnancy, severe renal failure and allergy to contrast) as described by Squizzato et al
- Line 324. Authors should consider including a brief commentary about the limitations related to not performing distal veins examination like when radiologists perform a comprehensive study : Thank you for that correction. Below you can find our addition to the manuscript along with the references to support this argument
Whether a full examination for distal DVT is appropriate or is still a debate, there are publications showing that 48% of patient with PE have contemporary isolated distal clots while this percentage for negative PE population is only 12% (Nchmi et al). Unfortunately these patients would be considered false negative while performing a standard POCUS protocol instead of a full CUS performed by a radiologist. ( ref:
Elias A, Mallard L, Elias M et al.: A single complete ultrasound investigation of the venous network for the diagnostic management of patients with a clinically suspected first episode of deep venous thrombosis of the lower limbs. Thromb. Haemost. 89, 221–227 (2003). 35. Schellong SM, Schwarz T, Halbritter K et al.: Complete compression ultrasonography of the leg veins as a single test for the diagnosis of deep vein thrombosis. Thromb. Haemost. 89, 228–234 (2003). 36. Stevens SM, Elliott CG, Chan KJ, Egger MJ, Ahmed KM: Withholding anticoagulation after a negative result on duplex ultrasonography for suspected symptomatic deep venous thrombosis. Ann. Intern. Med. 140, 985–991 (2004).
-
Nchmi A, Ghaye B, Noukoua CT, Dondelinger RF: Incidence and distribution of lower extremity deep venous thrombosis at indirect computed tomography venography in patients suspected of pulmonary embolism. Thromb. Haemost. 97(4), 566–572 (2007).
Reviewer 2 Report
In this paper, the authors elaborate on an interesting review about diagnosing deep venous thrombosis (DVT) by a simple ultrasound scan carried out in a few minutes at the patient's bedside. According to data showed by authors, the point-of-care Ultrasound (POCUS) on the inferior extremities allows to detect or discard venous thrombosis with high sensitivity by the scan of 2 or 3 specific points along the common femoral vein and popliteal vein. DVT is well known to be associated with local venous insufficiency and severe cardio-respiratory morbidity complications such as pulmonary embolism (PE). In contrast, the treatment with low weight molecular heparin prevents these complications effectively.
The article is well written and easy to understand and provides a good insight for better management of this important clinical issue.
DVT and PE are very frequent in patients at the hospital with other clinical or surgical problems non directly related to coagulopathy. PE has been shown as one of the causes for decompensation of chronic diseases such as COPD or heart failure of unknown cause, associated with higher mortality and length of hospital admission (1). The early diagnosis and treatment of DVT and PE may prevent a significant proportion of the complications related to hospitalisation. Besides, the ambulatory treatment of patients with PE with a low risk of medical complications, discarded by a simple scale, was associated with a low rate of complications and secondary effects related to anticoagulant treatment (2).
As the authors underlined, “Well-designed large studies are needed to evaluate whether D-dimer in conjunction to POCUS can increase diagnostic accuracy”. I wonder if the authors could go a step further in proposing the design of this “large study” using POCUS as a diagnostic tool for DVT, especially in all the hospitalised subjects, for instance, with Wells scale >=2 for DVT diagnosis. I suggest you elaborate on this point briefly in the discussion.
In the same vein and with the availability of simple ultrasound probes, it would be interesting to propose residency training programs to diagnose DVT in the same way they are trained in other conventional physical examination techniques.
Minor suggestions
I suggest you add some arrows to point out "echogenic thrombus" in Figures 4 and 5.
1-Aleva FE, Voets LWLM, Simons SO, de Mast Q, van der Ven AJAM, Heijdra YF. Prevalence and Localization of Pulmonary Embolism in Unexplained Acute Exacerbations of COPD: A Systematic Review and Meta-analysis. Chest. 2017 Mar;151(3):544-554. doi: 10.1016/j.chest.2016.07.034. Epub 2016 Aug 12. PMID: 27522956.
2- Roy PM, Penaloza A, Hugli O, Klok FA, Arnoux A, Elias A, Couturaud F, Joly LM, Lopez R, Faber LM, Daoud-Elias M, Planquette B, Bokobza J, Viglino D, Schmidt J, Juchet H, Mahe I, Mulder F, Bartiaux M, Cren R, Moumneh T, Quere I, Falvo N, Montaclair K, Douillet D, Steinier C, Hendriks SV, Benhamou Y, Szwebel TA, Pernod G, Dublanchet N, Lapebie FX, Javaud N, Ghuysen A, Sebbane M, Chatellier G, Meyer G, Jimenez D, Huisman MV, Sanchez O; HOME-PE Study Group. Triaging acute pulmonary embolism for home treatment by Hestia or simplified PESI criteria: the HOME-PE randomised trial. Eur Heart J. 2021 Aug 7:ehab373. doi: 10.1093/eurheartj/ehab373. Epub ahead of print. PMID: 34363386.
Author Response
Comment 1: I suggest you add some arrows to point out "echogenic thrombus" in Figures 4 and 5:
Thank you so much for that! Will do.
Comment 2: 1-Aleva FE, Voets LWLM, Simons SO, de Mast Q, van der Ven AJAM, Heijdra YF. Prevalence and Localization of Pulmonary Embolism in Unexplained Acute Exacerbations of COPD: A Systematic Review and Meta-analysis. Chest. 2017 Mar;151(3):544-554. doi: 10.1016/j.chest.2016.07.034. Epub 2016 Aug 12. PMID: 27522956.
Very interesting work. Will implement as indicated
Comment 3: Roy PM, Penaloza A, Hugli O, Klok FA, Arnoux A, Elias A, Couturaud F, Joly LM, Lopez R, Faber LM, Daoud-Elias M, Planquette B, Bokobza J, Viglino D, Schmidt J, Juchet H, Mahe I, Mulder F, Bartiaux M, Cren R, Moumneh T, Quere I, Falvo N, Montaclair K, Douillet D, Steinier C, Hendriks SV, Benhamou Y, Szwebel TA, Pernod G, Dublanchet N, Lapebie FX, Javaud N, Ghuysen A, Sebbane M, Chatellier G, Meyer G, Jimenez D, Huisman MV, Sanchez O; HOME-PE Study Group. Triaging acute pulmonary embolism for home treatment by Hestia or simplified PESI criteria: the HOME-PE randomised trial. Eur Heart J. 2021 Aug 7:ehab373. doi: 10.1093/eurheartj/ehab373. Epub ahead of print. PMID: 34363386.
Very interesting work. Will implement as indicated
Reviewer 3 Report
Interesting and well described work, well supported with clear discussion. I appreciated the synthetic style of work development that helps to remind presented data. Please add the following interesting article in the references: Mumoli N, Vitale J, Giorgi-Pierfranceschi M, Sabatini S, Tulino R, Cei M, Bucherini E, Bova C, Mastroiacovo D, Camaiti A, Palmiero G, Puccetti L, Dentali F; PRACTICUS Study Investigators. General Practitioner-Performed Compression Ultrasonography for Diagnosis of Deep Vein Thrombosis of the Leg: A Multicenter, Prospective Cohort Study. Ann Fam Med. 2017 Nov;15(6):535-539.
Author Response
Thank you so much for your valuable help. We appreciate your feedback and will implement your suggested references as indicated.
Mumoli N, Vitale J, Giorgi-Pierfranceschi M, Sabatini S, Tulino R, Cei M, Bucherini E, Bova C, Mastroiacovo D, Camaiti A, Palmiero G, Puccetti L, Dentali F; PRACTICUS Study Investigators. General Practitioner-Performed Compression Ultrasonography for Diagnosis of Deep Vein Thrombosis of the Leg: A Multicenter, Prospective Cohort Study. Ann Fam Med. 2017 Nov;15(6):535-539.